# Ella versus Simoa Serum Neurofilament Assessment to Monitor Treatment Response in Highly Active Multiple Sclerosis Patients

**DOI:** 10.3390/ijms232012361

**Published:** 2022-10-15

**Authors:** Martin Nötzel, Luise Ina Werder, Tjalf Ziemssen, Katja Akgün

**Affiliations:** Department of Neurology, Technische Universität Dresden, 01307 Dresden, Germany

**Keywords:** neurofilament, Ella, Simoa, method comparison, alemtuzumab

## Abstract

The measurement of serum neurofilament light chain (sNfL) is of growing importance in the field of neurology. In the management of multiple sclerosis, it can serve as a useful marker to assess disease activity and treatment response. This paper compares two available methods, namely the Single Molecule Array (Simoa) and the Ella microfluid platform, to measure longitudinal sNfL levels of 42 highly active multiple sclerosis patients treated with alemtuzumab over a period of 36 months. In order to assess the methods agreement, Bland–Altman plots and Passing–Bablok regression were analyzed. Here, we show that despite the fact that Ella measures around 24% higher values than Simoa, both are equally suitable for longitudinal sNfL monitoring.

## 1. Introduction

Neurofilaments are an essential component of the cytoskeleton in neurons; they rise after neuronal damage not only in cerebrospinal fluid (CSF), but also in blood, indicating neuroaxonal injury independent of the causal pathway [1].

In recent years, growing attention has been paid to serum neurofilament light chain (sNfL) as a reliable quantification and longitudinal follow-up evaluation marker of neuroaxonal damage. As a biomarker in clinical practice, it is suggested to use sNfL both to quantify disease activity and to monitor the treatment response in neurological diseases such as multiple sclerosis (MS) [2,3,4]. It is known that sNfL is increased in patients with relapsing remitting MS (RRMS) and correlates with MRI activity, disability, and relapse activity [5,6,7]. A number of papers have discussed neurofilament as a potential predictive marker for MS outcome and disease progression [8,9,10]. Previously, a distinct reduction in sNfL levels during immunomodulatory therapy and clinical and MRI response were observed in MS patients [6,11,12].

Since different ultrasensitive immunoassays to measure sNfL are available, the questions arises of whether these methods are interchangeable [13,14]. Two innovative technologies being used are the NF-light Advantage Kit for the Single Molecule Array (Simoa, Quanterix) and the Simple Plex Human NF-L Cartridge on the Ella microfluidic platform (Protein Simple) [15,16].

Based on patient data from highly active MS patients treated with the induction therapy alemtuzumab, we compared the two methods and investigated, for the first time, whether Ella and Simoa are equally suitable for longitudinal monitoring of sNfL level response to alemtuzumab treatment.

## 2. Results

### 2.1. Comparablity of sNfL Measurement Using Simoa Versus Ella

The global mean sNfL concentration measured by Ella (16.5 pg/mL, 95% CI [14.6, 18.4]) was significantly (*p* < 0.001) higher compared to Simoa measurement (12.5 pg/mL, 95% CI [10.8, 14.2]). Measurements with both platforms presented similar intra-assay CVs on Ella (6.8 pg/mL, 95% CI [6.4, 7.1]) and Simoa (4.6 pg/mL, 95% CI [4.4, 4.9]). In order to assess the comparability of the two methods, Bland–Altman plots (Figure 1A,B) as well as Passing–Bablok regression (Figure 1C) were analyzed. The mean difference in the Bland–Altman plots are presented either in pg/mL or in percentages, demonstrating a mean bias of 3.50 pg/mL (LOA [−4.07 pg/mL to 11.07 pg/mL]) or 24.00% (LOA [−33.30% to 81.31%]). Hence, Ella measured greater values compared to Simoa. Furthermore, the difference rises for higher mean values, indicating a positive trend proportional to the magnitude of the measured sNfL value. The Pearson correlation shows a weak positive correlation (r = 0.34) but a strong positive correlation (r = 0.70) for Bland–Altman plots in percentage and units, respectively.

The Passing–Bablok regression line has a slope of 1.64 and an intersection of −3 (Figure 1C). The approximate resulting regression equation is
sNfL_Ella_ = 1.6 × sNfL_Simoa_ − 3(1)

Considering the 95% CI, the regression line and the identity line (y = x) are significantly different, indicating a systematic difference between the methods. Since the influence of biological covariates cannot be excluded, we additionally analyzed different subpopulations (sex, age, and body mass index) as described above and present the data in Table A1. However, no relevant deviations could be identified.

### 2.2. Serum NfL Levels Are Associated with Alemtuzumab Treatment Response

We monitored the sNfL levels of 42 patients during alemtuzumab treatment (Figure 2, Table A2). At baseline start, sNfL levels were higher in patients with recent relapse and MRI activity (Ella: 35.1 pg/mL, 95% CI [14.5, 55.7]; Simoa: 27.1 pg/mL, 95% CI [8.8, 45.4]) before alemtuzumab initiation. After initial treatment start, the sNfL values rapidly dropped and reached a steady state after about six months. There is a clear offset between the curves of both methods, which can be explained by the mean difference stated above. Importantly, the qualitative course of sNfL levels over the treatment period can be shown equally between both methods. After first and second alemtuzumab treatment course, most of the patients responded clinically well: 85% were free of relapses, 83% were free of new T2 or gadolinium-enhanced MRI lesions, and 76% presented no EDSS progression (defined by confirmed EDSS progression; ≥1.0 point increase if EDSS baseline score was <5.5; ≥0.5 point increase if EDSS baseline score was ≥5.5). In addition, the cohort was subdivided into responders (Months 15–36: No relapse activity, no MRI activity) versus non-responders (Months 15–36: Relapse activity and/or MRI activity) (Table A3 and Table A4). There was no relevant correlation between EDSS and sNfL levels measured by either method. Responder patients presented higher sNfL levels at baseline compared to non-responders, reflecting more aggressive disease activity before treatment start. The initial sNfL decrease was higher in the responder patients.

## 3. Discussion

Monitoring of sNfL levels is increasingly important in neurological diseases in clinical practice [17]. Especially in highly active MS patients, markers are of great relevance to prove high disease activity and later to define treatment response. Here, we demonstrate that both the Simoa and Ella platforms are suitable for longitudinal sNfL measurement in MS patients and present reliable results in the lower as well as higher value range. Both methods demonstrate very good intra-assay CV, supported by automated duplicates or triplicates on the devices. Measurements were performed on the same day and under the same conditions to exclude confounders.

The results demonstrate that Simoa and Ella measurements do not agree. Ella shows about 24% higher values than Simoa. Although similar anti-NfL antibodies and blockers are used in both techniques, different calibrators are discussed as potential reasons for differences in sNfL levels [18]. 

Our data are in line with a previous study which showed that Ella overestimates Simoa’s results by about 17% [13]. In this study, a HD-1 analyzer compared to a HD-X analyzer was used. However, the company guarantees complete agreement of both analyzers. In contrast to the previous study, we present follow-up data of patients that demonstrated initially highly active disease followed by rapid response to alemtuzumab therapy. Moreover, the number of measured samples was appreciably higher. The results indicate a higher mean difference for greater sNfL values. The influence was small for our data, but it must be considered if much higher sNfL values are to be monitored. This aspect might be important for sNfL monitoring of other neurodegenerative diseases that are associated with even higher sNfL levels (>100 pg/mL), such as amyotrophic lateral sclerosis or dementia [18]. Taken together, Ella and Simoa are not interchangeable. Consequently, it is recommended to pay particular attention when interpreting results based on reference values. In fact, most researchers to date have utilized Simoa [17,19,20]. Passing–Bablok regression (Equation (1)) can be used to estimate sNfL values for Ella from Simoa.

We think that both methods are very helpful in the monitoring of neurological diseases, in addition to clinical examination and standard diagnostic tests. In MS, definition of disease activity and treatment response is important for individualizing patient care. Alemtuzumab is known as an induction therapy that induces T and B cell lysis by the monoclonal antibody against CD52, especially in highly active MS patients. The sNfL evaluation by both methods demonstrates correlation with clinical and subclinical disease activity markers in our cohort. Serum NfL levels rapidly decreased after the first alemtuzumab infusion cycle and remained at a low steady state level after initial treatment [6,11]. Since the platforms do not show the same absolute value, Ella and Simoa should not be interchanged while monitoring treatment response.

Both devices offer advantages that can be favorable for certain users. The lower limit of quantification (LLOQ) is 2.70 pg/mL for Ella and 0.174 pg/mL for Simoa [17,18]. For patient samples that are expected to have very small sNfL levels, Simoa might be more suitable. However, we assume that the LLOQ of Ella should be sufficient to address most clinical questions. Another advantage of Simoa is the active cooling, which allows its use in non-air-conditioned rooms. In addition, the assays are more flexible in use, as the entire cartridge does not have to be discarded after one measurement. Thus, individual measurements are possible in a cost-efficient manner. On the other hand, the Ella device has the advantage of a smaller size compared to the Simoa HD-X Analyzer. Furthermore, the overall costs for Ella measurements are generally lower.

A new assay was recently developed for the ADVIA Centaur XP immunoassay system from Siemens Healthineers. It shows very promising results and might become a good alternative to the methods presented here [21]. As far as we know, it is not yet commercially available.

In summary, Ella measures 24% higher sNfL values than Simoa. The difference depends on the measurement range and can be determined using a regression line. Moreover, both methods were equally able to show the qualitative course of sNfL for highly active multiple sclerosis patients treated with alemtuzumab. While Simoa is well suited for measuring particularly small sNfL values, Ella is more cost efficient for larger routine diagnostics.

## 4. Materials and Methods

### 4.1. Patient Samples

For this study, we analyzed the data collected from 42 patients diagnosed with highly active RRMS (Table 1). They received alemtuzumab treatment after critical review of clinical and MRI data as well as extensive discussion of available treatment options.

Throughout the first infusion course, 12 mg alemtuzumab was infused on five consecutive days. During the second course (12 months later), alemtuzumab was applied on three consecutive days. Serum samples were collected regularly over the treatment period of 36 months. Clinical parameters including relapse activity were recorded at three monthly visits. A cerebral MRI was performed every 12 months.

### 4.2. Ella and Simoa Serum Neurofilament Measurement

We used the Simple Plex Human NF-L Cartridge measured on an Ella instrument (Protein Simple, San Jose, CA, USA) and Simoa NF-light Advantage Kit for Simoa and measurement on a HD-X analyser (Quanterix Corp, Boston, MA, USA) to assess sNfL. After blood collection, serum samples were frozen and stored at −80 °C. For both devices, samples were measured directly after thawing according to the manufacturer’s recommendations on the same day and under identical laboratory conditions. A dilution of 1:2 and 1:4 was used for Ella and Simoa, respectively. Separate standard curves were generated prior to all measurements, and both high and low controls were included.

### 4.3. Statistical Analysis

The intra-assay coefficients of variation (CV) of manufacturer-provided controls were automatically calculated in duplicate (Simoa) or internal triplicate (Ella). Associated quantitative variables are presented as mean and 95% confidence interval (CI). Median sNfL values were compared by Wilcoxon rank test.

For Bland–Altman plots and Passing–Bablok regression, sNfL values differing by more than three standard deviations from the mean of the particular method were considered as outliers and were removed prior to statistical analysis. Furthermore, we included additional data of the same patients over a total period of 113 months. The acceptance interval for the Bland–Altman plot was defined to be 0 ± 1.96 × (CV_Ella^2^ + CV_Simoa^2^) × 1/2, as suggested in previous studies [22]. According to the data, the inter-assay CV is 10.4% and 8.1% for Ella and Simoa, respectively [23,24]. This results in an acceptance limit of 0 ± 25.8%. If the limits of agreement (LOA) within the Blant–Altman plots exceed the acceptance interval, the agreement of the methods is rejected. The analysis was carried out with Python 3.9.8.

## Figures and Tables

**Figure 1 ijms-23-12361-f001:**
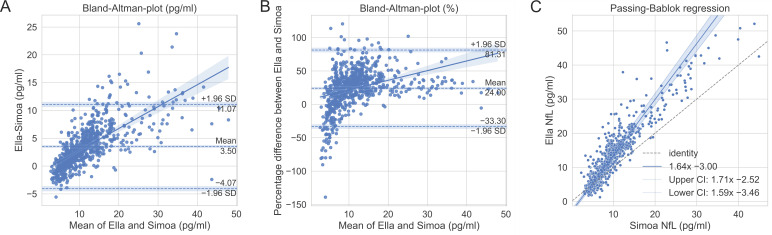
Comparison of Ella and Simoa methods. Bland–Altman plots in units (**A**) and percentage (**B**) with 95% confidence interval (CI) for mean, limit of agreement, and regression line. (**C**) Passing–Bablok regression with 95% CI. Dashed reference line is y = x, representing perfect method agreement.

**Figure 2 ijms-23-12361-f002:**
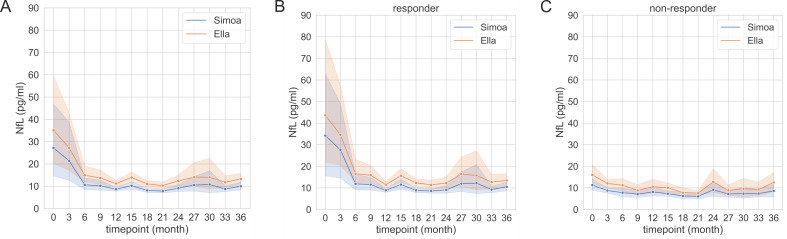
Timecourse of serum neurofilament light chain (sNfL) during alemtuzumab treatment in the whole cohort (**A**) in responders ((**B**), Months 15–36: No relapse activity, no MRI activity) and non-responders ((**C**), Months 15–36: Relapse activity and/or MRI activity), with 95% confidence intervals. The sNfL levels were measured using Ella and Simoa. (**A**: All patients (*n* = 42); (**B**) Responders (*n* = 29), (**C**): Non-responders (*n* = 13). Data time points are summarized quarterly.

**Table 1 ijms-23-12361-t001:** Descriptive statistics of sample population.

Total, N		42
Female, N (%)		32 (76)
Age in years, mean (SD)		35.7 ± 8.5
BMI, mean (SD)		24.2 ± 5.1
EDSS, mean (SD)		3.1 ± 1.3
Last pre-treatment, N (%)	None	3 (7)
	1st line	9 (21)
	2nd line	28 (68)
	Other	2 (5)
Patient with relapses in prior 1 year, N	0	12
	1	11
	2	11
	>3	8
New T2 or gadolinium-enhanced lesions in MRI in prior 1 year, N (%)	yes	29 (69)
	no	12 (29)
	unknown	1 (2)

1st line: Dimethyl-fumarate, Glatiramer acetate, Interferons, Teriflunomide; 2nd line: Natalizumab, Fingolimod, Siponimod, Daclizumab; Other: Privigen, Secukinumab.

## Data Availability

The data presented in this study are available upon request from the corresponding author.

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
