# Peer review of "Ella versus Simoa Serum Neurofilament Assessment to Monitor Treatment Response in Highly Active Multiple Sclerosis Patients"

_ijms, 2022, doi:10.3390/ijms232012361_

Round 1

Reviewer 1 Report

From my side, it is a great initiative to find better assay methods for sNfL. Since SIMOA is expensive and if sNfL has to be used across economic conditions the cost should be reduced. In that case, this might be the first step-

Few comments

-Did the authors check the temperature/storage-related variabilities as refrigeration facilities might not be the same in every part of the world

-Why authors did these two assays? There are other assays such as Umma. Is there any difference between this assay with SIMOA or E11a in terms of sensitivity? Which assay is better?

-Is there any study comparing SIMOA and E11a in CSF?

-A cross-comparison of the assays (in a table) would be interesting – Taking two assays results from this study and other from literature to see what are the values other got and this study found to get the cut-off value of each assay or the sensitivity of assays.

-While the main concern of sensitivity and cost-effectiveness vs benefit, authors can discuss more which method is best based on these points. Which method will be used in the diagnostic lab to detect sNfL?

-While a similar result was published a year before (https://pubmed.ncbi.nlm.nih.gov/33830650/) then what new is added in this study? Need to provide the highlight of the study that added new in the literature from this study.

-What is the main difference between the two assay systems needs to explain clearly

Author Response

We would like to thank the reviewers for their constructive comments.

Did the authors check the temperature/storage-related variabilities as refrigeration facilities might not be the same in every part of the world

Response:

The samples were stored at -80 degrees Celsius and measured directly after thawing as stated in the methods section. It has been shown before that this procedure should not affect the results (https://doi.org/10.1515/cclm-2022-0007).

Why authors did these two assays? There are other assays such as Umma. Is there any difference between this assay with SIMOA or E11a in terms of sensitivity? Which assay is better?

Response:

The two methods are well established in our lab and used on a regular basis for clinical diagnostics. Not many studies are available comparing sNfL levels of Simoa to other methods. A previous study showed Simoa to be superior to older methods for the sNfL level assessment such as ELISA and electrochemiluminescence immunoassays (https://doi.org/10.1515/cclm-2015-1195). In addition, a new NfL assay was recently introduced for the ADVIA Centaur XP immunoassay system (Siemens Healthineers) (https://doi.org/10.3389/fneur.2022.935382). The first results are promising and we added it our discussion (lines 143-146).

Is there any study comparing SIMOA and E11a in CSF?

Response:

As far as we know there is no earlier study directly comparing Simoa and Ella in CSF.

A cross-comparison of the assays (in a table) would be interesting – Taking two assays results from this study and other from literature to see what are the values other got and this study found to get the cut-off value of each assay or the sensitivity of assays.

Given the paucity of data on comparisons of sNfL assessment methods, we find it difficult to compile enough data for the suggested table. A general comparison of all available methods for the sNfL measurement would go beyond the scope of this paper.

While the main concern of sensitivity and cost-effectiveness vs benefit, authors can discuss more which method is best based on these points. Which method will be used in the diagnostic lab to detect sNfL?

What is the main difference between the two assay systems needs to explain clearly

Response:

One of the main differences of both methods is the lower limit of quantification achieved by Simoa. It is therefore possible to reliably measure sNfL values in the lower measurement range. The choice of one or the other device depends on the field of application. We think that for larger routine diagnostics Ella can be well used for most clinical settings. If someone's intention is to examine small sNfL levels, then Simoa might be the better choice. To clarify that, we added that fact to the summary (lines 146-148).

While a similar result was published a year before (https://pubmed.ncbi.nlm.nih.gov/33830650/) then what new is added in this study? Need to provide the highlight of the study that added new in the literature from this study.

We compared a higher number of samples (703 vs. 203) and we included a longitudinal study for patients treated with alemtuzumab. To reflect this, we have reworked the discussion accordingly (lines 111-113).

Reviewer 2 Report

The article by NÓ§tzel et al., titled “Ella versus Simoa serum neurofilament assessment to monitor treatment response in highly active multiple sclerosis patients” contains interesting perspectives of comparative analysis of neurofilament light chain (NfL) measurement methods in neurological diseases. The authors have shown in-depth knowledge of the subject. It is a well-planned study and the results were presented clearly which supports the conclusion. However, I have a few concerns which are listed below for the authors to improve this manuscript –

Minor Concerns-

1-    In figure 1 and 2 are not easy to understand as resolution is very low. The authors are suggested to enhance the clarity and enlarge the size of both figures.

Author Response

We would like to thank the reviewers for their constructive comments.

In figure 1 and 2 are not easy to understand as resolution is very low. The authors are suggested to enhance the clarity and enlarge the size of both figures.

Response:

We adjusted the ratio of the figures and increased the font size. Furthermore, the resolution of the figures is now much higher.